



# Values in water management

Erik Mostert

Department Water Management, Delft University of Technology, Stevinweg 1, 2628 CN Delft, the Netherlands

*Correspondence to*: Erik Mostert (e.mostert@tudelft.nl)

**Abstract.** Values play a key role in water management and natural resources management more generally. They can be used by individuals, groups, organisations and whole societies to judge and justify actions. In addition, they can be enacted in practice and embodied in water institutions and infrastructure. Values can be seen as responses to the necessities of human existence and the need to live together and form functioning societies. Mechanisms that can explain their origin, effects and development include reciprocity, socialisation, formal and informal sanctioning, new information, and socio–economic

change. Depending on the type of value, different measurement methods can be used, including surveys, interviews, focus groups, Q methodology, content analysis of cultural texts such as newspaper articles, argumentation analysis and participant observation. The article recommends research at the level of social groups and organisations because of the large role these play in water management. Consequently, it is important to measure their values, including both their environmental values – how they relate to their environment – and their social values – how they relate to each other.

**1 Introduction**

Values play a key role in water management (e.g. Groenfeldt, 2013). They can be described as "principles or standards of behaviour; one's judgement of what is important in life" (Stevenson, 2010). Without sufficient knowledge of the prevalent values, it is impossible to understand how humans interact with their environment and with each other.

In a recent article in this journal, Roobavannan et al. (2018) reviewed the role of values in socio–hydrological models. To

improve the generalisability and predictive power of these models, they proposed the use of Values–Beliefs–Norms (VBN) theory (Stern et al., 1999; Stern, 2002). VBN theory explains pro-environmental behaviour, such as environmental activism and support for environmental policies, by pro-environmental personal norms. These norms are explained in turn by personal values and by personal beliefs that the realisation of these values is threatened and that the individual can help to alleviate the threat.

VBN theory has been used to explain for example climate change mitigation and adaptation measures at farm level (Sanderson and Curtis, 2016) and support of farmers and non-farmers for water policies (Sanderson et al., 2017). However, like all theories, it has its limitations. It has not been developed to explain collective behaviour, such as the construction of reservoirs, and assumes that values are relatively stable. This makes it less suitable for explaining for instance the "pendulum shift" in many basins from water resources development and exploitation to protection and restoration (Van Emmerik et al.,



2014; Chen et al., 2016; Mostert, 2018a). VBN theory moreover does not consider enforcement of values and norms. The main research method for applying VBN theory is survey research, but survey research is not always possible or practical, e.g. for past periods. And finally, the distinction between "values" and "norms" is problematic since they can range from very fundamental to very applied and do not fit neatly in two categories.

The aim of this article is to develop a conceptual theory of values that can inform future water management research. This theory should help explain collective as well as individual behaviour, account for changes in values, and consider the social and cultural aspects of values. In addition, this article provides an overview of methods for actually measuring values. The article focuses on the values of human actors have, rather than the value or values they assign to water. Consequently, valuation methods will not be discussed in any detail.

The structure of this article is simple. After presenting the methodology in section 2, the different types of values and their effects and origins are discussed in section 3. In section 4, an overview of the main methods for measuring values is given. In section 5 the question which types of values are most important in water management is addressed and recommendations for future research are given. The article closes with a short conclusion.

## 2. Methodology

The main research method used for this article was literature study. First, literature on VBN theory and work of Schwartz (1992, 1994, 2006 and 2009), to which this literature refers, was read. The work of Schwarz could be contrasted with different cultural approaches (Thompson et al., 1990; Hofstede, 1991) and sociological and political science literature on community and reciprocity, most notably the work by Robert Putnam (e.g. Putnam et al., 1994; Putnam, 2001). In addition, literature on ideology (e.g. Mannheim, 1960), discourse analysis (e.g. Fairclough, 2003) and framing (e.g. Scheufele, 1999)

influenced the analysis. The literature study was complemented by literature on the Theory of Planned Behaviour (Ajzen, 1991, 2011; White et al., 2009) and various other literatures, referred to in the next sections.

    The aim of the literature study was not to review any strand of literature exhaustively, but to identify and eventually integrate the major insights and approaches from different strands of literature. Terminological differences were not a central concern. Instead, I looked for substantive differences and similarities, in particular for elements missing from the literature already

sudied that could be used as building blocks for the conceptual theory being developed.

    The main database used was Google Scholar because of its comprehensiveness. The algorithm Google Scholar uses for ranking results is not public, but it is clear that citation counts and the occurrence of key words in the title are very important (Beel and Gipp, 2009; Martin–Martin et al., 2017). This makes Google Scholar appropriate for finding seminal articles. The search terms used varied and were the result of a trial−and−error process aiming for highly relevant results while limiting the

number of irrelevant results. To find newer articles as well, some searches were limited to literature published in the last five years and in a few cases all publications citing an article of special interest were scanned.





For making an overview of measuring methods use could be made of the general literature on research methodology. The main task was to apply the insights from this literature to the issue of values. In addition, references to examples from the field of socio–hydrology and water management research more generally were included.

## 3. Values: what they are and how they function

### 3.1 Types of values

As mentioned in the introduction, this article uses the term "value" primarily to refer to principles or standards of behaviour and judgements of what is important in life. Values in this sense are inherently human since only human actors have principles and standards of behaviour and make judgements.[1] Things can have a value too, but these are assigned values, assigned by human actors applying their own principles, standards and judgements. They can be valued for the specific

goods and services they provide ("instrumental value"), they can be valued for their own sake ("intrinsic value") or both. For example, a wetland can be valued because it attenuates flooding but also for its beauty and peacefulness. In the first case it could be replaced by flood defences, in the second case it cannot.[2]

Values in the sense of human principles and standards can range from fundamental and general, such as beauty and peace, to very applied and specific, such as a prohibition to drain a specific wetland. All these will be called "values." The term

"norm" will be used when only applied and specific values are referred to.

Values can be used to judge and justify behaviour; and they can be enacted in practice and embodied in institutions and artefacts, such as water laws and water management infrastructure. Following a hint in Schwartz (1992), these two types of values will be called ideal values and actual values respectively. Ideal values are often referred to in public discussions and policy documents, while actual values motivate actual behaviour and can best explain this.

Ideal and actual values may be either individual or cultural. Individual values are the principles and standards of behaviour that individuals use or enact, while cultural values are the values that groups, organisations or society use or enact. They may differ significantly. In public discussions, individuals may not be free to express their values, there may be pressure to

---

[1] Some higher developed animals have values too, not abstract principles and standards but concrete moral emotions that influence their behaviour: Bekoff and Pierce (2009).

[2] Often, "intrinsic value" is used to refer to the value that nature or animals have independent from humans (see McShane, 2007). This idea has been criticised not only because values are essentially human, but also because it artificially separates humans and their environment, and separation − of culture and nature, rationality and emotions, men and women − results in different forms of domination, in exploitation and in environmental degradation (Mellor, 2007). To overcome separation, a new category of values has been proposed, relational values, which reside neither in humans nor in their environment, but in their relations (Chan, 2016; Muraca, 2016). Yet, while these relations are of crucial importance, humans can be distinguished from their environment, and if the relation is valued, this is done by the humans.



conform or, conversely, discussions may get polarised (cf. Wutich et al., 2010). Policy documents of an organisation do not necessarily reflect the values of all or even the majority of its members, but primarily those of the drafters and decision−makers. Likewise, artefacts and institutions reflect primarily the values of the most influential individuals.

When similar, the cultural values may reflect the individual values, but the opposite may also be true. Individuals develop
many of their values early in life as a result of socialisation, and throughout their life they are influenced by the mass media and different forms of monitoring and control (see section 3.3). When induced to act against their own values, they may modify their values to reduce cognitive dissonance (cf. Mills, 1958; Felson, 2014). Moreover, they may adopt the cultural values of the groups they identify with (White et al., 2009; Fielding and Hornsey, 2016). In addition, organisations may attract and select only individuals that share their pre-existing cultural values.

The relation between ideal and actual values varies. Individuals may act upon their ideal values or adjust their ideal values to their actual behaviour to reduce cognitive dissonance. In both cases their ideal and actual values will be similar. They will differ when individuals lack the means to act upon their ideal values and when in specific cases conflicting ideal values are activated (see section 3.2). Likewise, individuals, groups, organisations and whole societies may act upon their ideal values, adjust their ideal values, or act differently. Yet, a large difference between ideal and actual values may create problems. Even
very powerful actors often feel the need to legitimise their actions in order to minimise opposition and secure their position (cf. Majone, 1989), and the larger the discrepancy between their words and deeds, the harder this will be.

The relation between values and facts is complex. The position taken in this article is that values and facts can be analytically distinguished but not separated. Values can be grounded in the necessities of human existence (Schwartz, 1994) and the need to live together and form functioning societies (Hofstede, 1991; cf. Bekoff and Pierce, 2009). They provide answers to a
number of fundamental social tensions, such as the role of the individual versus the collectivity, equality versus hierarchy, tradition versus change, and security versus development (Thompson et al., 1990; Hofstede, 1991; Schwartz, 1994, 2006). Taking these tensions as a starting point, different typologies of values can be developed, such as the one presented in Table 1.

**Table 1: A typology of values (Schwartz, 2006), based on three dimensions: embeddedness versus intellectual and affective autonomy, hierarchy versus egalitarianism, and mastery versus harmony. Values are compatible with values of an adjacent type (1 and 7 are adjacent too) but conflict with values of the opposite type.**

| Types of values | Examples |
| --- | --- |
| 1. Harmony | Unity with nature, environmental protection, beauty, peace |
| 2. Embeddedness | Order, obedience, tradition, security, politeness |
| 3. Hierarchy | Authority, humbleness, wealth, power |
| 4. Mastery | Ambition, daring, influence, success, recognition, setting own goals |
| 5. Affective autonomy | Pleasure, excitement, variety |
| 6. Intellectual autonomy | Broadmindedness, curiosity, creativity, freedom |



| 7. Egalitarianism | Justice, equality, honesty, loyalty, responsibility, accept portion in life |
| --- | --- |

The values listed in Table 1 can be linked directly with water management issues. We may hypothesise that actors who attach much importance to harmony will assign an intrinsic value to water and the others only an instrumental value. Embeddedness and hierarchy values will lead to a preference for regulatory approaches to water management, ambition and

mastery values to a preference for economic approaches, and egalitarian values to a preference for participatory approaches. Furthermore, mastery and affective autonomy values will be more likely to engender conflicts, and harmony and egalitarian values solidarity and cooperation.

### 3.2 Mechanisms

Values can influence behaviour in different ways. In VBN theory, behaviour is motivated by fundamental individual values,
mediated by individual beliefs that these values are under threat and the individual can help to alleviate the threat. In addition, behaviour is influenced by individual capabilities, such as knowledge, skills and finances, and by contextual factors, such as community expectations, advertising, government regulation and financial incentives (Stern, 2002).

As briefly mentioned by Stern et al. (1999) and Stern (2002), individual values can also have a direct effect on behaviour independent of beliefs. This will happen when individuals have internalised the value and compliance gives positive feelings
and non-compliance negative feelings, e.g. feelings of guilt.

Cultural values and especially cultural norms can influence behaviour in other ways as well. If enacted in practice, they can inform actors what is appropriate, effective and socially acceptable in a specific situation ("descriptive norms": White et al., 2009; Minato et al., 2010). Compliance with these norms may be unattractive in itself, but if others reciprocate and comply as well, the benefits of compliance may outweigh the costs (section 3.3).

Cultural values can also be enforced. Compliance can be controlled by the parties interested in the specific values (Ostrom, 1990) or in the social order in general. Sanctions may include reputational damage and social exclusion (e.g. Yu et al., 2017). Compliance with legal norms can in addition be controlled by the police and the courts, who may impose fines or imprison perpetrators. The effectiveness of enforcement has been found to depend on the (perceived) probability, severity and swiftness of the sanction and on individual characteristics of the addressees, such as the degree self–control. In addition, the
social context is important (Worrall et al., 2014). In small communities, for instance, non-compliance is detected more readily and informal sanctions can be applied more easily and have more serious consequences than in large anonymous societies.

Cultural values can also influence behaviour via the physical artefacts and institutions that embody these values. Artefacts and institutions can be modified and used for different purposes than originally intended (cf. Cleaver, 2002), but they are
often long–lasting and can influence the water management practice long after the values on which they were once based have changed. The Dutch polders provide clear examples of this. They were developed in the Mediaeval period primarily to





increase agricultural production (Van de Ven, 2004), but they still dominate the Dutch landscape now that recreational and environmental values have become important as well.

When in a specific case different values potentially apply, a key question is which are activated and which not. Values can be activated by verbal as well as visual cues. For instance, talk of money stimulates individualistic values and pictures of authority figures may stimulate respect for authority (Kahneman, 2011; cf. Vatn, 2009). These cues selectively "frame" the issues involved (Scheufele and Iyengar, 2012).

When more than one value is activated, actors can try to balance the different values. This assumes these values are commensurable, that is, measurable by the same standard, such as their contribution to economic development or "utility" (Thacher and Rein, 2004). When different actors support different values, politics may come into play. One value may gain ascendency until the negative consequences for another value become too severe to ignore, which may then dominate for a while ("cycling": Thacher and Rein, 2004). This mechanism may explain the "pendulum shift" mentioned in the introduction. Alternatively, the different actors may become engaged in a social learning process that involves recognition of all relevant values and a joint search for solutions that satisfy all these values as much as possible (Mostert et al., 2007; Pahl-Wostl et al., 2007). Institutional solutions to prevent that some values are overlooked completely include sectoral legislation and separate authorities protecting these values ("firewalls": Thacher and Rein; the "separation principle": Mostert, 2015a).

### 3.3 Origin and development

Values can be explained in different ways. Two explanations were already briefly mentioned in the previous section: patterns of behaviour that become "descriptive norms" and reciprocity. When actors interact repeatedly, they may cooperate in order to increase the chance that others will reciprocate their behaviour ("direct reciprocity": Axelrod, 1984; Nowak, 2006). In the absence of repeated interaction, they may cooperate in order to improve their reputation and thereby increase the chance that others will cooperate with them ("indirect" or "generalised reciprocity": Nowak, 2006; Putnam et al., 1994). Generalised reciprocity is most effective in communities with many horizontal social connections ("social capital": Putnam, 2001). Social connections entail mutual obligations, as when individual A does individual B a favour. When social connections are widespread, A may do B a favour without expecting anything in return from B, but from other community members. Widespread social networks reduce possibilities for opportunism as this would threaten the reputation of the person involved. They increase trust and foster the development of norms of cooperation (Putnam, 2001). These norms are essential because without them opportunism would not be viewed negatively.

Cooperation can also be stimulated by community values, such as solidarity. Solidarity refers to both mutual support in furthering a common interest and support for needy community members (Keessen et al., 2016). While there is often an element of reciprocity in solidarity, the main motivation is not self–interest in a narrow sense, but a sense of being similar to others, of having a common identity and constituting a "we" instead of only separate "I's" (Tönnies and Loomes, 1963; Selznick, 1994). For sustaining a sense of community, it is important that people feel they matter to the community and the community to them, and that their emotional and other needs are fulfilled (cf. McMillan and Chavis, 1986).



Values can be internalised early in life via a process called "socialisation", involving a combination of conditioning, learning by example, explanation and discovery (Selznick, 1994; Elsenbroich and Gilbert, 2014). Throughout life actors are influenced by the media and media framing (e.g. Scheufele, 1999). In addition, values are internalised via "disciplining" (Foucault, 1975). Disciplining involves monitoring the behaviour and performance of individuals, using for instance

statistical data and examinations; comparing their performance to a norm; ranking them; and applying graduated positive or negative sanctions, such as prizes, marks, promotion or degradation.

Some values are officially enacted or adopted. This applies especially to legal rules (with the exception of rules of customary law) and the ideal values expressed in official policy documents.

Values can change as a result of new information (e.g. Minato et al., 2010). Many values have a factual element and can be

justified in terms of the contribution they make to realising more fundamental values. In VBN theory, new information can change personal beliefs, which in turn can change personal behavioural norms. This is, however, not a straightforward process. For example, people may agree that the past few years have been exceptionally hot and dry, but they may not agree whether this is a sign of climate change or just climate variability and if climate change, what its cause is and what can be done about it. The vast majority obtain their information on climate change and climate change research from the media.

Whether they believe this information and pay attention to it at all depends on their worldviews and fundamental values and on the trust they have in the media and science (Weber, 2010, 2016; cf. Wynne, 1992). And with the advent of the social media and the algorithms these media use, people are increasingly exposed to information that confirms worldviews and values (Amrollahi and McBride, 2019).

Values can also change as a result of socio–economic change. Socio–economic change may result not only in the growth of

specific groups with specific values (Roobavannan et al., 2017), but also in a change of their values (Mostert, 2018a). According to Human Development theory (Welzel et al., 2003), economic growth increases the resources and possibilities of individuals, which in turn results in a shift from traditional survival values to self–expression values and in pressure for more political freedom and democracy. The developments in many Western countries after the Second World War offer support for this theory. In this period, the economy grew significantly, the average level of education increased, traditional bonds

relaxed, and the authority of the old elites was increasingly questioned, especially by young people. This led in 1968 to large-scale student revolts in Paris, Berlin and Berkeley (Righart, 1998).

Changes of values often involve discussion. This may range from small-scale dialogues and large-scale "megalogues" (Etzioni, 1996), to heated debates involving rhetorical tricks, misinformation and threats. Discussion may also take to the streets in the form of demonstrations that show the numerical strength and determination of the protestors. Violence may

break out and revolutions may be started. This is very far removed from open dialogue, but it has to be recognised that discussion always reflect power relations that influence who can speak when and how much weight their arguments carry (e.g. Fiske, 1990; Bourdieu, 1991).

**Table 2: An overview of the conceptual theory of values**





*Types of values:*

- Values of human actors (focus in this article) and values they assign to things (not discussed in detail)

- From fundamental to applied (the latter also called " norms")

- From formal (explicit and officially adopted) to informal

- Ideal values, used to judge and justify behaviour, and actual values that are enacted in practice and embodied in institutions and artefacts

- Individual values, used or enacted by individuals, and cultural values, used in or enacted by a group, organisation or society at large

*Mechanisms:*

- Means: applied values are means for reaching more fundamental values

- End: values are ends in themselves; observing them may give positive feelings and not observing them negative feelings

- By example: actual values inform individuals what is appropriate, effective and socially acceptable

- Reciprocity: values are observed to stimulate, and as long as, others do the same

- Informal sanctioning, e.g. reputational damage and social exclusion

- Formal sanctioning of formal norms, e.g. fines and imprisonment

- Influence via long-lasting institutions and artefacts

Values can be selectively activated; some may (temporarily) dominate over others; different values may be combined in individual cases as a result of social learning; and protection of different values may be delegated to separate institutions.

*Origin and development:*

- Practices turning into actual values

- Direct and indirect reciprocity

- Sense of community

- Socialisation early on in life

- Influence of the media throughout life

- Disciplining

- Explicit design and adoption

- New information

- Socio–economic change

- Discussions, from open dialogue to manipulation of public opinion

- Protests and revolutions

Different factors may be at work simultaneously.





Table 2 summarises the conceptual theory presented in this section. Not all elements of this theory will be important in every case, but they can function as so-called "sensitising concepts" that give a general sense of reference and guidance and suggest directions along which to look (Blumer, 1954). In addition, they can be used selectively as building blocks for simpler theories or models that can be tested empirically. However, the simpler they are, the more limited their field of
5    application generally should be (cf. Levins, 1966; Troy et al., 2015).

## 4. Measuring values

To do any research on the role of values in water management, it is essential to measure them, quantitatively or qualitatively. The methods that can be used differ for individual and cultural values and for ideal and actual values (Table 3). As this article focuses on the values of human actors rather than the value they assign to things, valuation methods will not be
10    discussed in detail (see on this Jacobs, 2018).

**Table 3: Major methods for measuring different types of values**

| | Individual | Cultural |
|---|---|---|
| Ideal | <ul><li>Surveys</li><li>Interviewing</li><li>Q methodology</li><li>Content analysis of egodocuments</li></ul> | <ul><li>Surveys[a]</li><li>Interviewing[a]</li><li>Q methodology[a]</li><li>Content analysis of cultural texts (newspapers, social media, etc.) using coding</li><li>Argumentation analysis</li><li>Focus groups</li><li>(Participant) observation</li></ul> |
| Actual | <ul><li>Inference from observed or reported individual behaviour and artefacts, using for instance interviews and questionnaires</li><li>Behavioural experiments</li></ul> | <ul><li>Inference from observed or reported collective behaviour and cultural artefacts, using for instance interviews, questionnaires and participant observation</li><li>Behavioural experiments</li></ul> |

a: Provided individual and cultural values are similar

### 4.1. Individual ideal values

15    A common method for measuring individual ideal values is survey research. Survey research has been used in socio–hydrology to obtain factual information not otherwise available and gauge experiences and perceptions (e.g. Ferdous et al.,



2018, Massuel et al., 2018). To assess values, questions can be included, such as "how important in your life is it to protect nature" or "how important in your life is it to become rich" (1: not important at all; 5: extremely important). Survey research is not without its pitfalls (Kelley et al., 2003). The exact wording of the questions can have a large impact on the responses, there may be translation issues, and response may be low and not representative. Moreover, specific expertise and much time

are required, and unless one is willing and able to repeat the survey, it is impossible to measure change.

An alternative for original survey research is to use data from existing surveys, such as the World Values Survey (www.worldvaluessurvey.org) and the European Social Survey (www.europeansocialsurvey.org) (e.g. Bjørnskov, 2006). The first data for the World Values Survey were collected in 1981–1982, covering ten countries only, but in the last completed "wave" (2010–2014) sixty countries were covered. The data are national. Data from the European Social Survey

are available at a disaggregated level as well, but for most countries the units used are Eurostat's statistical regions (the "NUTS" regions) instead of hydrologically relevant units, and the sample size per region is rather small.

A third option is to conduct new survey research using a tested questionnaire or questions from a tested questionnaire, such as the Schwartz Value Survey or the Portrait Values Questionnaire (Schwartz, 2006, 2009; Sandy et al., 2017; see for an example Schulz et al., 2018). This requires less time and expertise than completely original case study research, but much

more than using data from existing surveys.

In addition to survey research, interviews with open questions can be held. Questions may include for instance what the important water issues are for the interviewee and why these are important. As in surveys, the exact wording of the questions is crucial. Moreover, there is a large risk that interviewees give socially desirable answers.

Of each interview a report should be made. These can be analysed using a procedure called coding (Merriam and Tisdell,

2016). The first step in coding is the development of a set of "codes": relevant themes or topics that are mentioned or hinted at in the interviews. The codes can be based on a relevant theory, such as the theory presented in the previous section; they can be identified inductively, during the research; and they can be a mix of both (see for examples Mostert, 2015b, Bark et al., 2016, and Haeffner et al., 2018). The second step is to assign codes to relevant segments of the interviews reports. Thirdly, overviews can be made of the coded segments and patterns may emerge, for instance differences between

interviewees from different areas or between young and old interviewees. These patterns can then be analysed further.

A third method for measuring individual ideal values is Q methodology. Q methodology consists of the following steps (Van Exel and De Graaf, 2005):

1.    Collection of statements about the issue at hand (the "concourse"), using documents and interviews

2.    Selection of the 25 to 40 most relevant statements (the "Q set")

3.    Selection of respondents (the "P set")

4.    Ranking of statements by the respondents according to how much they agree with each, e.g. from 1 (agree least) to 5 (agree most), using a fixed distribution over the different categories (the "Q sorting")

5.    Statistical analysis of the resulting Q sorts, resulting in a limited number of factors that can explain most of the observed variety



6.       Interpretation of the factors, focusing on the statements that have very different average scores

Q methodology can be used with a limited number of respondents and results in an overview of different subjective views (the interpreted factors). It is not possible to quantify the frequency of the different views. Examples of the use of Q methodology in water management research include Raadgever et al. (2008), Vugteveen et al. (2010) and Forrester et al. (2015).

Fourthly, individual ideal values may be extracted from egodocuments, such memoirs, diaries and private letters, provided, of course, that such documents are available. Yet, egodocuments are not purely individual ( on Greyerz, 2010). Many are meant to be read, sooner or later, by others, and the authors are inevitably influenced by cultural factors, including cultural values. The main method for analysing egodocuments is content analysis, discussed below.

## 4.2. Cultural ideal values

Surveys, interviews and Q methodology can also be used for measuring cultural ideal values, but only if these values are similar to the individual ideal values that are effectively measured (see section 3.1). If they are not, or if this is uncertain, a better option is to measure the cultural values directly, using for instance content analysis of cultural texts.

Content analysis covers several techniques, such as discourse analysis, framing analysis and narrative analysis (Metag, 2016). It usually consists of three steps (Fairclough, 2003). The first step is the identification and, if necessary, selection of relevant "texts", such as newspaper articles (Wei et al., 2017), children's books (Mostert, 2015b), parliamentary proceedings (Mostert, 2018b), online comments (Bark et al., 2016), and photos (De Kruif, 2009). The second step is the identification of key themes in the texts. The third step an analysis of how these themes are covered. For these two steps coding as described in the previous section can be used.

Several computer programmes can support content analysis of large corpora of texts. They can all help in organising, visualising and retrieving data (Duff and Séror, 2005; see for examples Bark et al., 2016 and Haeffner et al., 2018). Some can also code texts automatically (Grimmer and Stewart, 2013; Castelfranchi, 2017). The simplest form involves the development by the researcher of a dictionary with keywords for each code. For the code "solidarity", keywords could include "community", "unity", "mutual support", "cohesion" and of course "solidarity". The programme then uses this dictionary to code the texts automatically. A more complex form of automatic coding involves supervised learning. Researchers first code some 100 to 500 texts manually and input the results in the programme. The programme then "learns" how to code itself, based on the occurrence and frequency of words that are typical for each code. Some programmes can also help in the identification of the codes themselves, based on the occurrence and frequency of words and (in the case of Structural Topic Models) metadata such as the age and gender of the author (Grimmer and Stewart, 2013; Roberts et al., 2015).

In addition to coding, texts can be analysed in terms of argumentation (Van Eemeren et. al., 2014). Arguments in their simplest form consist of a conclusion or "claim" and premises on which the conclusion rests. For instance, a policy plan may claim that a specific hydropower dam should be built because this will result in less $CO_2$ emission. Premises are often



implicit, e.g. in this case the premises that 1) reducing $CO_2$ emissions is positive; 2) there are no viable alternative for reducing $CO_2$ emissions, such as energy saving; 3) the dam will not lead to more energy use; and 4) there are either no negative side-effects or they are less important than the benefits. Each of these premises can be analysed as the conclusion of a prior argument resting on prior premises. Some of these premises will be factual, some will be values, and many will have

a mixed character.

A third method for measuring cultural ideal values directly is focus group research. A focus group is a small group that discusses a topic determined by the researcher, who also moderates. The discussions do not necessarily reflect individual ideal values as there may be pressure to conform, participants may be censored, or conversely positions may get polarised (Wutich et al., 2010). From a cultural point of view these mechanisms are highly relevant, and to the extent that the focus

group is representative of the larger group or organisation of interest, the discussions reflect cultural ideal values.

A fourth method is participant observation (Jorgensen, 1989). This method can be used to find out which arguments are used in specific settings and which values have currency, provided the presence of the researcher does not influence the discussions too much. If participant observation is not possible, the researcher may interview individual members of the group or organisation of interest and ask which arguments are used in specific situations, but this will be second-hand

information.

### 4.3. Actual values

Actual values cannot be measured directly, but they can be inferred from individual or collective behaviour, institutions and artefacts. There is, however, the risk of circular reasoning, for example when one uses the construction of a sewage treatment plant to infer pro-environmental values and then uses the pro-environmental values to explain the construction of the sewage

treatment plant. The major advantage of inference is that no significant bodies of text are needed, so this method can be used in the study of pre-historic civilizations (e.g. Pande and Ertsen, 2014).

Behaviour and artefacts can be observed directly, during field visits or using participant observation; they can be derived from for instance election results and membership figures of social organisations; and they can be self-reported, e.g. in interviews or in responses to surveys that ask about activities. A well–known example of the use of membership figures and

survey data to infer community values is Robert Putnam's book "Bowling alone: The collapse and revival of American community" (Putnam, 2001).

Actual values might also be inferred from behavioural experiments, such as the degree of altruism shown in different types of games (Park et al., 2017). A major advantage of behavioural experiments is that by changing conditions, behaviour can change and explanations for the actual values can be developed. A major limitation is that the experimental settings can

differ significantly from the real−life settings of interest.



## 5. Discussion

The overview of values presented in this article is not the first overview that tries to integrate different approaches. In 2017, Schulz et al. developed a conceptual framework that distinguishes between fundamental values, governance−related values and assigned values or water values (Schulz et al, 2017; see also Schulz et al., 2018). In terms of this article, fundamental
values are the ideal values of individuals and groups that may or may not be enacted upon, governance−related values are the actual values reflected or embodied in processes, institutions and interpersonal or intergroup relations, and assigned values are the values assigned to water. Terminological differences apart, this article is much more specific concerning the different causes and effects of values and it adds an overview of different ways to measure values.

Attention to values does not remove the need for hydrological and economic research and for political analysis. Values can
support the peaceful co-existence of humans with each other and with their environment, but only if the hydrological constraints are respected and basic needs, such as food and shelter, are met. Moreover, there is always the question whose values prevail and who will benefit. In socio–hydrological models, values may even be ignored completely, provided the values are stable. In these cases hydrological change and societal response can be linked directly. As soon as values change, however, the relation between hydrological change and societal response will change as well (cf. Srinivasan et al., 2017).
Depending on the type of values, different measurement methods can be used. The choice of method should depend on practical considerations, such as the availability of data sources, but also on the type of values that is most relevant. For water management research actual values seem more relevant than ideal values because they are linked directly with actual behaviour, but ideal values are important too. They play an important role in ethical and political discussions that can influence societal response. On top of this, there are good data sources available, such as the World Values Survey and the
European Social Survey.

The issue whether to measure individual values or cultural values is a complex one. Individuals cannot be understood fully if we ignore the cultural context in which they were brought up and function, but the cultural context cannot be understood fully if we ignore the individuals that participate in, support and influence it. Approaches that start from individuals and their values may explain the formation and functioning of primary groups satisfactory, but not the formation and functioning of
complex governance structures. Starting from the highest level of "society", however, may not be very useful for understanding the internal dynamics of society and the role of individuals.

The approach proposed here is to start research at the intermediate level of social groups and organisations and move up or down when necessary for understanding the broader social context, the internal dynamics of the groups and organisations, and the role of individuals. Social groups and organisations are actors in their own right. They emerge or are created, they
grow and shrink and can amalgamate, their tasks and character can change, and eventually they disappear or are abolished. Moreover, they act and interact. In the short term their actions and interactions are influenced by the dominant cultural values, but in the longer term they also change these values. Similarly, they are influenced by but also change their (water) environment. To complicate matters, those changing the environment often differ from those experiencing the consequences





and there may be a time lag between change and consequences. How these differences and time lags are handled depends on, among others, the changing cultural values.

## 6. Conclusion

This article has discussed the role of values in water management. It has discussed different types of values, their effects and origins, and the different methods that can be used to measure them. It has not discussed each type, effect, origin and method exhaustively, but it has provided a general overview that can inform water management research. A major conclusion of the article is the importance of different social groups and organisations in water management. Hence, it is important to measure their values, both their environmental values – how they relate to their environment – and their social values – how they relate to each other.

*Data availability.* No data sets were used in this article.

*Competing interests.* The author declares that he has no conflict of interests.

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
