# Peer review of "Values in water management"

_Hydrology and Earth System Sciences, 2019_

## Referee Comment (RC1) · Anonymous Referee #1 · 23 Jul 2019

This paper discussed the role of values in water resources management. The author conducted a literature study on the conceptual frameworks and mechanisms of values. Then the author discussed the methods to measure values in water management. A major conclusion of the paper is that different social groups and organizations are crucial actors in water management and it is important to measure their values, including both environment values and social values. The article is on a topic of interest to the audience of Hydrology and Earth System Sciences. The author effectively addressed most of the reviewers' comments of previous submission. I have the following comments that I hope the author could address in the revision.

Specific comments:

1. After reading the manuscript, I have a similar impression as the previous referee #1 that I think the paper is more like an opinion paper or a review paper instead of a

research paper.

2. The abstract should clearly describe the methodology and main findings of the study.

3. Page 2, Line 5: I think this study is more about to provide an overview of concepts of values, instead of developing a conceptual theory. I would suggest to revise this statement.

4. Page 2, Lines 7-8: The study focuses on the values in general, rather than the value human assign to water. This choice of study focus might make the paper not relevant enough to the fields of hydrology and water resources management. Also, the title of the paper should be changed if this is the case.

5. Page 13, Lines 12-13: The statement about values in socio-hydrological models is not accurate since many studies have tried to implement values in socio-hydrological models. I would suggest the author to extend this paragraph and provide a more comprehensive discussion about values in socio-hydrological models or water management models. This would make the article more relevant to HESS.

―――――――――――――――――――

---

## Referee Comment (RC2) · Anonymous Referee #2 · 24 Jul 2019

I found this article to be well written, interesting, with a clear structure and persuasive arguments. Much of the work discussed in this article was unknown to me and I expect to be unknown to most water readers - in this sense I think it could be a useful review type paper. However, most prominently it doesn't really talk all that much about water. I am persuaded that value theory and its methods might be useful, but very little of the article discusses how it has been/could be applied to water management. I think more needs to be done to link these ideas to water if this work could constitute a HESS review or research article. For example, I see in the references that many cited works are water-based studies, can the author discuss the water relevant specifics of these in more detail? Could water management inspired examples for Table 1 and 2 be given?

---

## Referee Comment (RC3) · Francesca Pianosi (Referee) · 26 Jul 2019

I think the topic is potentially very interesting and useful in a community where these topics are often overlooked or poorly known.

However, in my opinion the manuscript does not fully achieve its goal, because the discussion is too broad and not really tailored to the water management field. One could replace the string "water management" by "climate change adaptation" or "ecosystems conservation" or "landscape management" etc., and most of the paper would read equally well. For example, Table 1 and 2, and relevant discussion: these are all very generic concepts - how do they apply in water management? Can the author gives examples and discuss them?

Similarly, while many references are indeed from water-related fields, their content is not discussed and contrasted in any detail, so one can attach them to the particular

field only because of their titles, not the way they are discussed. For example, P. 11 L. 3: "Example of the use of Q methodology in water management research include ... Forrester et al. (2015)". What's the value of simply listing these references? What I would have expected here is a discussion of how the Q methodology was used in those studies, what they achieved, what limitations were found when applying the method to the water management field, etc.

Because of this lack of specificity, I also found it difficult to fully appreciate the paper conclusions. For example:

P. 13 L. 16: "For water management research actual values seem more relevant than ideal values because they are linked directly with actual behaviour, but ideal values are important too." Where is this coming from? Is it the author's opinion or is it a conclusion from the literature? Where was this discussed?

P. 13 L. 27: "The approach proposed here is to start research at the intermediate level of social groups and organisations and move up or down when necessary for understanding the broader social context, the internal dynamics of the groups and organisations, and the role of individuals. Social groups and organisations are actors in their own right." Again, what experiences / previous findings led the author to propose such approach? Can you give some examples of how it would work in practice (which are the social groups and organisations to be involved, what internal dynamics would be investigated, etc.)?

P. 14 L. 6: "A major conclusion of the article is the importance of different social groups and organisations in water management." What evidence backs this statement? Where was it presented in the review? I might have missed it, but if so, I suppose other readers may do as well?

So in conclusion I think the manuscript is very interesting in principle, but it needs substantial revisions to include concrete examples from water management and more tailored discussion, in order to become a useful contribution to HESS.

---

## Author Comment (AC1) · 22 Aug 2019

First of all, I would like to thank the reviewers for their comments.

Possibly the most fundamental comment is that the paper "doesn't really talk all that much about water" (reviewer 2) and is "not really tailored to the water management field" (Francesca Pianosi). Most of paper would apply equally to for instance "climate change adaptation", "ecosystems conservation" or "landscape management" (Francesca Pianosi). The reviewers would like to see more discussion of how the different concepts apply in water management and more examples (reviewers 2, Francesca Pianosi). They would like to see more discussion of the contents of the references from water-related fields, discussing for example the experiences with the use of Q methodology (Francesca Pianosi) or how values have been included in socio-hydrological models (reviewer 1).

[Figure]

In response, I would argue that the whole paper deals with water, but not exclusively and always explicitly. How humans interact with water depends on the prevalent values, but these values do not depend exclusively on water. Hence, to understand human-water interactions, it is necessary to look beyond water. The same can be said of for instance climate change adaptation, ecosystems conservation or landscape management. Much of the paper is equally relevant for these issues, but that in itself does not reduce the paper's relevance for water management and hydrology.

That being said, it is essential that the paper shows how the different concepts apply in water management. Here I see several possibilities for improvement. I can provide more examples and give more illustrations from the field of water.

It would be interesting to review the experiences with Q methodology concerning water issues, but that would require a paper on its own. With respect to measuring techniques, the present paper only aims to provide an overview. The purpose of including references to applications of Q methodology without discussing the applications is simply to inform readers that Q methodology has been applied to water issues and to suggest further readings for those interested in the details. There is currently no need to review the role of values in socio-hydrological models since the review by Roobavannan et al. from 2018 is still up to date.

Additional comments reviewer 1

Reviewer 1 wondered whether the article is relevant enough for hydrology and water management since it focuses on the values of human actors instead of the value they assign to water (valuation). I would argue it is. Water management is a human activity and hence human values are very important. The examples and illustrations I plan to add should make this clearer. Valuation is of course highly relevant too, but this deserves an article or articles on its own. I do refer to a good recent overview of valuation techniques. Given the broad scope of my article, the broad title "Values in water management" seems appropriate; for an article focusing on valuation a better

title would be "The values of water".

Reviewer 1 mentioned that the paper is more like an opinion paper or a review paper than a research paper and questioned the use of the term "conceptual theory". These issues have been discussed before in the review process. The paper is based on literature research. It reviews a lot of literature but it is not a "classical" review paper, and it contains a discussion section but it is not a pure discussion paper. While I think the emphasis is on research, I accept that the paper has a somewhat hybrid character. The term "conceptual theory" best describes the content of the paper, especially of sections 3 and 4, which distinguish different types of values and discusses their causes and effects. I recognise that the term "theory" may be used differently in different disciplines. That is an issue I cannot resolve.

Reviewer 1 also commented that the abstract should clearly describe the methodology and main findings of the study. I will add the main research method used. The main findings have already been included.

Moreover, reviewer 1 commented that the statement on Page 13, Lines 12-13 is not accurate since many studies have tried to implement values in socio-hydrological models. I will try to make the statement clearer. Literally, I did not state that values are ignored completely in socio-hydrological models values, but that they "may be ignored", provided they are stable. In that case, and only in that case, ignoring values does not affect the performance of the models negatively.

Additional comment reviewer 2

Reviewer 2 would like to see water management inspired examples for Table 1 and 2. In these tables there is very little room to give examples. As discussed, I plan to give more examples, but this will be mostly in the main body of the text.

Additional comments Francesca Pianosi

Francesca Pianosi had several comments concerning the support for the conclusions

and proposals. First, concerning the statement that "for water management research actual values seem more relevant than ideal values because they are linked directly with actual behaviour, but ideal values are important too." The relevance of actual values for water management research follows directly from the definition of actual values as values that are enacted in practice or embodied in institutions and artefacts. They "motivate actual behaviour and can best explain this" (section 3.1). The relevance of ideal values is explained on P. 13, L. 18 and further.

Secondly, what led the author to propose starting research at the intermediate level of social groups and organisations and how would this work in practice (which social groups and organisations to be involved, what internal dynamics to investigate, etc.)? In fact, the proposal is based on the preceding paragraph, which discusses the limitations of research that starts at the individual level or at the highest level of "society". Which social groups and organisations and which internal dynamics to investigate is indeed a crucial question. The simplest answer is: the most important groups, organisations and dynamics. Sometimes, most or all may already be known at the start of the research, but in other cases these should become clear during the research. I will develop the proposal more.

Thirdly, what evidence backs the conclusion of the article concerning the importance of different social groups and organisations in water management? I agree this conclusion needs to be supported more explicitly. It is based, first, on the importance in water management of collective behaviour, such as the construction of reservoirs or flood protection works. This involves groups and organisations and not individuals acting on their own. But secondly, even when individuals act on their own, their individual values that motivate their actions are influenced by cultural values, so by the dominant values in the groups and organisation they are a member of.

In conclusion, I plan to make the following improvements:

1. Most importantly: add examples and illustration of how the different concepts discussed in the paper apply to water management and hydrology.

2. Add in the abstract that the paper is based on literature study.

3. Clarify the statement that in socio-hydrological models values may be ignored provided they are stable.

4. Develop the proposal to start research at the intermediate level of social groups and organisations.

5. Provide more explicit support for the conclusion concerning the importance of social groups and organisations in water management.

—————————————————